# Social aloofness is associated with non-social explore-exploit decisions
Evan Knep[1], Xinyuan Yan [2], Cathy S. Chen[2], Suma Jacob[2], David P. Darrow [3], R. Becket Ebitz[4,5], Nicola Grissom [1,5] ✉ & Alexander B. Herman [2,5]

How humans resolve the explore-exploit dilemma in decision making is central to how we flexibly interact with both social and non-social aspects of dynamic environments. However, how individual differences in the cognitive computations underlying exploration relate to social and non-social psychological flexibility traits remains unclear. To test this, we probed decision-making strategies in a cognitive flexibility task, a restless three-armed bandit task, and examined how individual differences in cognitive strategy related to social and non-social traits measured by the Broad Autism Phenotype Questionnaire (BAPQ), a well-validated, clinically-relevant, community instrument, in a large (N = 1001) online sample. In contrast to prior links found between exploratory behavior and cognitive rigidity, we found that differences in choice behavior and exploration were primarily associated with social phenotypes as captured by the BAPQ aloof subscale. Higher scores on the BAPQ aloof subscale, indicative of reduced social interest and engagement, were associated with decreased shift rates, increased win-stay/lose-shift behavior, heightened sensitivity to negative outcomes, and reduced exploration. Reinforcement learning (RL) modeling further revealed that reduced exploration in high aloof individuals was driven by lower decision noise rather than increased cognitive rigidity, suggesting that decreased exploratory behavior may reflect a reduced tendency for stochastic exploration rather than an inflexible learning process. Sparse canonical correlation analysis reveals that the strongest loading for these non-social reward-related measures are in fact socially coded items. These results suggest that differences in motivation to seek information, especially in social contexts, may manifest as decreased exploratory behavior in a non-social decision-making task. Our findings additionally highlight the potential for using computational approaches to reveal general cognitive mechanisms underlying social functioning.

The explore–exploit tradeoff is a fundamental component of adaptive behavior, determining whether an agent seeks novel information (exploration) or capitalizes on prior rewards (exploitation). This balance is critical for decision-making across diverse contexts, shaping how individuals flexibly navigate uncertainty[1,2]. Explore–exploit behaviors are predominantly studied in non-social environments, where agents must balance exploration and exploitation in resource foraging, economic decisions, and learning strategies. However, these behaviors also play a crucial role in social interactions. In social contexts, individuals must decide whether to engage with new partners or rely on existing relationships, influencing trust formation, cooperation, and social network dynamics[3]. Expanding social

connections may provide access to novel resources, whereas maintaining established relationships ensures stability.

Decision-making tasks, especially bandit tasks, have proven effective at measuring explore–exploit balance in laboratory and ecological settings[4–8]. Bandit tasks require participants to repeatedly choose between options with uncertain and often changing reward probabilities, forcing them to balance the short-term benefits of exploiting known rewards with the potential long-term gains of exploring less familiar options. By tracking how individuals adapt their choices in response to changing reward contingencies, these tasks offer a computationally precise method for assessing decision-making flexibility. However, it remains unclear what aspects of real-world

[1]Department of Psychology, University of Minnesota, Minneapolis, MN, USA. [2]Department of Psychiatry, University of Minnesota, Minneapolis, MN, USA. [3]Department of Neurosurgery, University of Minnesota, Minneapolis, MN, USA. [4]Department of Neurosciences, Université de Montréal, Montreal, QC, Canada. [5]These authors contributed equally: R. Becket Ebitz, Nicola Grissom, Alexander B. Herman. ✉e-mail: ngrissom@umn.edu

functioning are accessed by laboratory explore–exploit measures such as these. As these tasks typically involve sequences of repetitive choices in a single-agent setting, they are often argued to primarily reflect cognitive flexibility or rigidity, potentially failing to capture the social flexibility component of explore-exploit behavior.

Yet these tasks may go beyond measuring surface-level cognitive traits, with evidence suggesting that they often tap into deeper neurocomputational mechanisms that underlie more complex behaviors[9]. For example, it has long been understood that the same underlying cognitive mechanisms that facilitate flexible decision-making in non-social contexts also play a crucial role in social interactions[10,11]. If we can leverage explore–exploit paradigms to access social flexibility, we can better understand the foundational cognitive processes driving both social and non-social flexibility, offering a more comprehensive understanding of adaptive behavior in complex environments[12–17].

In this study, we aimed to examine how explore–exploit balance in a restless bandit task relates to social and non-social flexibility in a large online sample ($n = 1001$). To asses distinct components of flexibility, we used the Broad Autism Phenotype Questionnaire (BAPQ), a well-validated self-report survey designed to capture dimensional variation in social and non-social autism-related traits in the general population[18–21]. The BAPQ includes three subscales: social aloofness, behavioral rigidity, and pragmatic language, each measuring different aspects of cognitive and social adaptability. We focused on the aloof and rigid subscales, as both directly relate to patterns of engagement with dynamic environments and may reflect distinct forms of flexibility. Importantly, although the BAPQ originates in autism research, our goal was not to study subclinical autism, but to leverage the BAPQ as a dimensional tool to parse social disengagement (aloofness) and cognitive inflexibility (rigidity) as potential drivers of explore–exploit behaviors in the broader population.

The social aloofness subscale captures an individual's tendency to withdraw or disengage from social interactions, making it highly relevant to social flexibility. Individuals high in aloofness often show reduced social motivation[22], which could influence exploration tendencies, particularly in uncertain or changing environments. If exploration in decision-making reflects an underlying drive for engagement with new opportunities, more aloof individuals may be less inclined to seek novel options, favoring a more exploitative approach that prioritizes known rewards. This aligns with findings that reduced social motivation is linked to behavioral patterns of social avoidance[23] and lower sensitivity to changing reward contingencies[24]. However, it remains unclear whether this tendency generalizes beyond social settings to influence exploration in non-social decision-making contexts.

The rigidity scale, on the other hand, measures inflexibility in thought and behavior, including resistance to change and a strong preference for routine. This subscale directly aligns with cognitive and behavioral adaptability in non-social contexts, making it particularly well-suited for testing relationships between decision-making flexibility and explore–exploit behaviors. A more rigid individual may struggle to adjust to changing reward contingencies, leading to a greater reliance on previously learned strategies rather than exploring new possibilities. However, given the established correlation between rigidity and aloofness[25], it is possible that exploitative individuals exhibit both tendencies, with aloofness contributing to reduced engagement in novel experiences and rigidity reinforcing reliance on established routines.

By exploring these relationships, we aim to determine whether explore–exploit laboratory paradigms, such as the restless bandit task, capture both social and non-social flexibility. If both aloofness and rigidity reflect a more exploitative strategy, it would suggest that these tasks reflect a broader profile of cognitive and social inflexibility, with shared mechanisms driving disengagement from novelty across domains. Alternatively, if these traits show distinct relationships with explore–exploit behavior, it may indicate that social and non-social flexibility are supported by separate processes, with exploration in decision-making reflecting different underlying motivations depending on the context. Understanding these

connections will help clarify whether laboratory paradigms measuring explore-exploit tradeoffs provide meaningful insights into flexibility beyond non-social decision-making, offering a more comprehensive framework for studying adaptive behavior in dynamic environments.

## Methods
### Data collection
All experimental procedures were consistent with and approved by the Institutional Review Board of the University of Minnesota (Study 00008486). 1001 online participants were recruited through Prolific, a recruiting tool for online experiments. Participation criteria included (1) being at least 18 years of age and (2) all participants must complete the task through desktop computers. These criteria were set to allow for a diverse general population from which to sample, with the device limitation implemented to mitigate any potential differences that could arise due to variation in modality used to complete the task. Participants were asked which gender identity they most identified with (Male, Female, Other). The gender identity distribution was 493 female participants, 496 male participants, and 12 participants who identified as other. Participant ages ranged from 18 to 75+, with an age distribution as follows. 18–24: 473, 25–34: 339, 35–44: 114, 45–54: 50, 55–64: 18, 65–74: 6, 75+: 1. All participants provided written informed consent. Participants received $3.10 if they completed the entire study, $3.50 if they had high accuracy during the task, and an additional $2.00 for responding to survey questions.

### Self-administered assessments
Participants were asked to complete the Broad Autism Phenotype Questionnaire (BAPQ)[26] as well as supply demographic information including sex assigned at birth, level of education, and household income.

### Exploration–exploitation paradigm
Exploratory behaviors were measured using a three-arm restless bandit task[27,28]. Each trial, participants were presented with three choices, each of which was associated with a reward probability that changed randomly and independently over time. Rewarded trials resulted in participants earning 1 point. The probabilistic nature of the task necessitates exploratory behaviors to monitor changes in reward rates as the task progresses, while encouraging exploitation of a target when the chance of reward is high. Participants were initially screened with a tutorial section, in which they needed to demonstrate basic task proficiency over 15 practice trials prior to beginning the final task. Each participant's restless "walk", or the volatility of the reward probability across 300 trials, was dictated by predetermined parameters for the likelihood of change in reward probability each trial (hazard rate) and the subsequent magnitude of change in reward probability (step size). In this experiment, we used a hazard rate of 0.6667 and a step size of ±0.2, constrained within the range of [0.1–0.9].

### Preregistration
This research was conducted without prior registration of the study design, hypothesis, or analysis plan.

### Data analysis
**General analytical techniques.** Data analysis was performed with custom PYTHON scripts. Shapiro–Wilks test was performed on relevant variables, with key variables such as BAPQ subscale scores and percent explore/exploit found to be non-normally distributed. The relationship among and between BAPQ scores, explore–exploit metrics, and other variables was subsequently tested with Spearman's correlation tests. All tests assumed an alpha of 0.05. No participants were excluded from our final data analysis.

**Punishment sensitivity.** To determine whether shift behaviors were reward driven, i.e. participants only shift when the previous trial was not rewarded, we used a measure of punishment sensitivity to assess each participant's relative shift probability following rewarded or unrewarded

trials. If shift behavior is not reward-driven, we will see punishment sensitivity close to zero.

$$\text{punishment sensitivity} = (\text{p(shift|loss)} - \text{p(shift|win)}/\text{p(shift)}) \quad (1)$$

**Hidden Markov model (HMM).** In order to examine how much an individual explored in the task, we fit a hidden Markov model (HMM) to the choice sequence to infer the latent explore/exploit state[5,29,30]. The HMM modeled exploration and exploitation as two latent goal states underlying choices. Each state is defined by a different emission matrix, i.e., the probability of making each choice under each state. When the model is fit to the choice sequence of a subject, it estimates a transition matrix that dictates the probability of transitioning from one state to another over time. Since the HMM assumes a Markovian process, states are time-dependent. The transition matrix is a mapping of past and future states, which describes the 1-time-step probability of transition between every combination of states.

In our model, there were four possible states (an exploit state for each choice, and one explore state). During each exploit state, the probability of choosing the exploited choice is 1, and 0 for other choices (fixed emission matrix). Exploration is modeled as a uniform distribution over choices because the uniform distribution over choices is the maximum entropy distribution of categorical variables. To accurately estimate the parameters of the model with a limited number of trials, the parameters were tied across exploit states such that each exploit state has the same probability of keep exploiting or begin exploring. Transitions out of exploration into exploit states were also tied. The model also assumed that subjects had to go through an explore state in between exploit states, even for a single trial exploration. The model estimates two unique free parameters—the probability of transitioning from exploration to exploitation and the probability of transitioning from exploitation to exploration.

To find the optimized transition matrix for each subject, we fit the model via expectation maximization using the Baum–Welch algorithm[31]. The algorithm was reseeded 10 times to avoid local maxima and find the global maxima. With the HMM transition matrix optimized from subject choice sequences, we then used the Viterbi algorithm to decode latent states from choices, allowing us to label each choice as either exploratory or exploitative.

Hidden Markov models (HMMs) have been identified as an especially effective method for capturing these explore–exploit behaviors across species[5,30,32,33]. By providing a framework for inferring explore–exploit behaviors and how individuals balance them under varying degrees of uncertainty, the HMM offers a quantitative measure of cognitive flexibility in decision-making contexts.

**Sparse canonical correlation analysis.** We applied sparse canonical correlation analysis (sCCA), a well-established and popular method to find associations across multiple sets of multivariate data. The main goal of sCCA is to find pairs of linear combinations that would maximize the correlations between two datasets with a sparseness parameter control, and how many dimensions are required for those correlations[34]. The pairs of linear combinations represent canonical variables, and the correlation between them is defined as canonical correlations.

In the current study, we have two datasets: BAPQ single-items ($B$) and exploration–exploitation indices ($E$).

$$B = \begin{pmatrix} B_{11} & \cdots & B_{1p} \\ \vdots & \ddots & \vdots \\ B_{n1} & \cdots & B_{np} \end{pmatrix} E = \begin{pmatrix} E_{11} & \cdots & E_{1q} \\ \vdots & \ddots & \vdots \\ E_{n1} & \cdots & E_{nq} \end{pmatrix} \quad (2)$$

where $n$ represents the number of participants, $p$ indicates the number of single items of BAPQ, $q$ indicates the behavioral indices from the exploration and exploitation task.

We aimed to find pairs of linear combinations to maximize the correlation between $B$ and $E$.

Mathematically, we can have linear combinations like:

$$\mathbf{X} = \mathbf{B}x \quad \mathbf{Y} = \mathbf{E}y$$

and we have

$$\text{Var}(\mathbf{X}) = x^{\text{T}} \sum_{bb} x$$

$$\text{Var}(\mathbf{Y}) = y^{\text{T}} \sum_{ee} y$$

$$\text{Cov}(\mathbf{X}, \mathbf{Y}) = x^{\text{T}} \sum_{be} y$$

Our aim is

$$\text{Max}_{x,y} x^{\text{T}} \sum_{be} y \text{ subject to constraints} ||x||^2 \leq 1, \ ||y||^2 \leq 1$$

$$P_1(x) \leq c_1, \ P_2(y) \leq c_2$$

where the $P_1(x)$ and $P_2(y)$ are lasso penalty functions (i.e., L1 regularization), and $c_1$, $c_2$ should satisfy:

$$1 \leq c_1 \leq \sqrt{p}, 1 \leq c_2 \leq \sqrt{q}$$

The values of $c_1$ and $c_2$ are chosen by $K$-fold cross-validation (CV), where the corresponding penalty values are chosen by grid search in increments of 0.1 between 0.1 and 1.0 to identify the combinations of parameters to maximize the Cov ($Bx$, $Ey$). As previous studies[35] did, we have 10 randomly resampled datasets as replication datasets, each of which consisted of two-thirds of the dataset with the full sample. We conducted sCCA in a predictive framework[36] to enhance the generalizability of the model. The L1 penalty for the single-item BAPQ dataset and behavioral indices from the exploration-exploitation task were tuned by 10-fold cross-validation, with a discovery ($n = 667$) and replication sample ($n = 334$). We obtained the best model with a penalty level of 1 on the single-item BAPQ and 0.9 on the exploration-exploitation behavioral indices. Analyses were implemented in R package PMA[37], available at https://rdrr.io/cran/PMA/man/PMA-package.html.

**Reinforcement learning (RL) models.** To model decision-making processes in the restless bandit task, we applied two reinforcement learning models: a standard delta-rule RL model and an RL model incorporating a choice kernel (RLCK).

The standard delta-rule RL model assumes that participants learn by updating value estimates ($Q$-values) for each option over time based on reward outcomes. The $Q$-value for a given choice is updated according to the reward prediction error (RPE):

$$Q_{t+1}^k = Q_t^k + \alpha(r_t - Q_t^k)$$

where $r_t$ is the received reward, $Q_t^k$ is the expected value of option $k$ at trial $t$, and $\alpha$ is the learning rate, which determines how strongly new information influences the updated value.

Choice selection was modeled using a Softmax probability function:

$$p(a_{t+1} = k) = \frac{e^{\beta Q_t^k}}{\sum_j e^{\beta Q_t^j}}$$

where $\beta$ is the inverse temperature parameter, controlling decision noise by determining the balance between random exploration and exploitation of higher-value options.

The RLCK model extends the standard RL framework by incorporating a choice kernel (CK), which accounts for an individual's tendency to repeat previous choices independent of reward history. The CK value for option kk is updated similarly to the $Q$-value update rule:

$$CK_{t+1}^k = CK_t^k + \alpha_C(a_t^k - CK_t^k)$$

where $a_t^k$ is a binary indicator of whether option $k$ was chosen on trial $t$, and $\alpha_C$ represents the rate at which choice persistence is updated.

The probability of selecting an option at trial $t$ is then determined by a Softmax function incorporating both $Q$-values and the choice kernel:

$$p_t^k = \frac{e^{\beta(Q_t^k + CK_t^k)}}{\sum_j e^{\beta(Q_t^j + CK_t^j)}}$$

This model accounts for both value-driven decision-making and habitual choice biases, allowing for a more nuanced characterization of explore–exploit behavior. Based on model comparison, the RLCK model provided a better fit to the data than the standard RL model, suggesting that choice persistence played a significant role in participants' decision strategies.

Model agreement was assessed using negative log-likelihood (NLL), which quantifies how well the model predicts observed choices. A lower NLL indicates better model fit, as it reflects a higher probability of the model assigning correct choices. Parameters $(\alpha, \beta, \alpha_C)$ were optimized using maximum-likelihood estimation (MLE) via the truncated Newton (TNC) method, with 30 replications per participant to avoid local minima.

**Permutation test**. Permutation testing was adopted to assess the significance of canonical variates[36]. We constructed a null distribution for all canonical components by shuffling the rows of the BAPQ single items and holding the behavioral indices vector constant. Thus, the linkage between BAPQ single items and participants' behavioral indices has been broken. We then conducted sCCA using the same regularization parameters on the realigned dataset to get canonical variates. If the canonical variates from the preserved sample are significant than the canonical variates from the permutated sample, we selected these canonical variates for further analysis. We performed permutations 5000 times and applied false discovery rate (FDR) to control for type I error due to multiple comparisons. The significance level is set at 0.05.

## Results

1001 participants over the age of 18 were recruited using the online platform Prolific to examine the relationship between social and non-social flexibility and latent cognitive processes underlying decision-making. To dissociate social and non-social cognitive phenotypes, participants were asked to complete the broad autism phenotype questionnaire (BAPQ) prior to the task[26]. The BAPQ is a commonly used tool to quantify cognitive traits across three major phenotypic domains, including aloofness, rigidity, and pragmatism (Fig. 1C). The total BAPQ score ranges from 36 to 216, a sum of the 12–72 score from each subscale, with higher scores associated with more severe autism-related phenotypes.

To examine individual differences in value-based decision-making strategies, we employed a restless three-armed bandit task. On each trial, participants were given three decks of cards to choose from, each of which was associated with a reward probability that changed randomly and independently over time (Fig. 1A, B). Participants accrued one point for every rewarded trial. The goal of the task is to maximize the number of points obtained over the duration of the experiment (300 trials). The dynamic reward contingency encourages participants to be flexible in their decisions, as the current best option may become worse in the future. To maximize reward, participants must exploit a favorable option when it is found while flexibly exploring alternatives to gather information. To verify that participants understood the task, we calculated the probability of obtaining a reward compared to the probability of reward if choosing randomly (chance). The results suggest that participants were performing the task significantly better than chance ($t(1000) = 79.14$, $p < 0.001$, Cohen's $d = 2.50$, 95% CI = [0.125, 0.131]) (Fig. 1D).

### High aloofness individuals exhibit decreased shift behaviors

One way to measure cognitive flexibility is to examine shift behavior[38,39]. We calculated the probability of shifting to a different option on a given trial and examined whether this shifting behavior correlated with BAPQ total or subscale scores. We found that the probability of shifting away from a previous choice was correlated with the BAPQ aloof subscale, with higher aloof scores reflecting decreased shift behavior (Spearman correlation (df = 999): rho = $-0.12$, 95% CI = [$-0.179$, $-0.054$], $p < 0.001$) (Fig. 1E). Despite its theoretical relevance to cognitive inflexibility, the BAPQ rigid subscale was not significantly associated with shifting behavior (Spearman correlation (df = 999): rho = 0.01, 95% CI = [$-0.052$, 0.071], $p = 0.789$), nor was the pragmatic subscale (Spearman correlation (df = 999): rho = 0.05, 95% CI = [$-0.007$, 0.116], $p = 0.101$) (Fig. S3). These results suggest that variability in choice flexibility is related to social aloofness, in particular, highlighting a potential link between social disengagement and reduced exploration in non-social decision-making contexts.

### Aloofness-related stay behaviors are outcome dependent

To determine if shift behavior was outcome-dependent, we examined measures of win-stay (repeating a choice following reward), lose-shift (shifting to a different choice following no reward), and a measure of punishment sensitivity that considers the ratio of shifts as a result of no reward relative to reward while controlling for overall amount of shift behavior (Eq. (1)). We find a positive correlation between BAPQ aloof scores and win-stay (Spearman correlation (df = 999): rho = 0.11, 95% CI = [0.044, 0.172], $p < 0.001$), as well as a negative correlation between BAPQ aloof scores and lose-shift (Spearman correlation (df = 999): rho = $-0.09$, 95% CI = [$-0.152$, $-0.026$], $p = 0.004$) behaviors, consistent with decreased shift behaviors with increases in aloofness (Fig. 1F, G). We also find a positive correlation between punishment sensitivity and BAPQ aloof (Spearman correlation (df = 999): rho = 0.09, 95% CI = [0.030, 0.151], $p = 0.004$), indicating an increased sensitivity to no reward relative to reward as aloofness increases (Fig. 1H). This suggests that high aloofness individuals showed decreased shift behaviors due to higher sensitivity to no reward, rather than a general strategy of increased choice repetition.

### Aloofness is correlated with decreased frequency of exploratory behaviors

Previous studies have shown that win-stay and lose-shift behaviors are not constant throughout the session; rather, they are elevated only during periods of exploration[5,30]. Therefore, we ask whether changes in outcome-dependent behaviors in high aloofness individuals are due to changes in exploratory strategy over time. To infer when exploration happens or how much individuals explore, we adopted a hidden Markov model (HMM) that models exploration and exploitation as two latent goal states underlying choices. Previous studies have found that labeling explore-exploit states using an HMM consistently and accurately modeled choice behavior across species[5,7,30]. To examine how much individuals explore, we calculated the frequency of exploratory choices as labeled by the HMM. This analysis revealed that aloofness was correlated with exploratory behaviors, with high BAPQ aloof scores reflecting decreased probability of exploring on a given trial (Spearman correlation (df = 999): rho = $-0.13$, 95% CI = [$-0.200$, $-0.069$], $p < 0.001$) (Fig. 2B).

Decreased exploratory choices could result from decreased frequency of exploratory bouts and/or shorter exploratory bouts. We therefore examined the transition probabilities between states, i.e., how likely an individual was to stop exploiting and start exploring, continue exploiting,

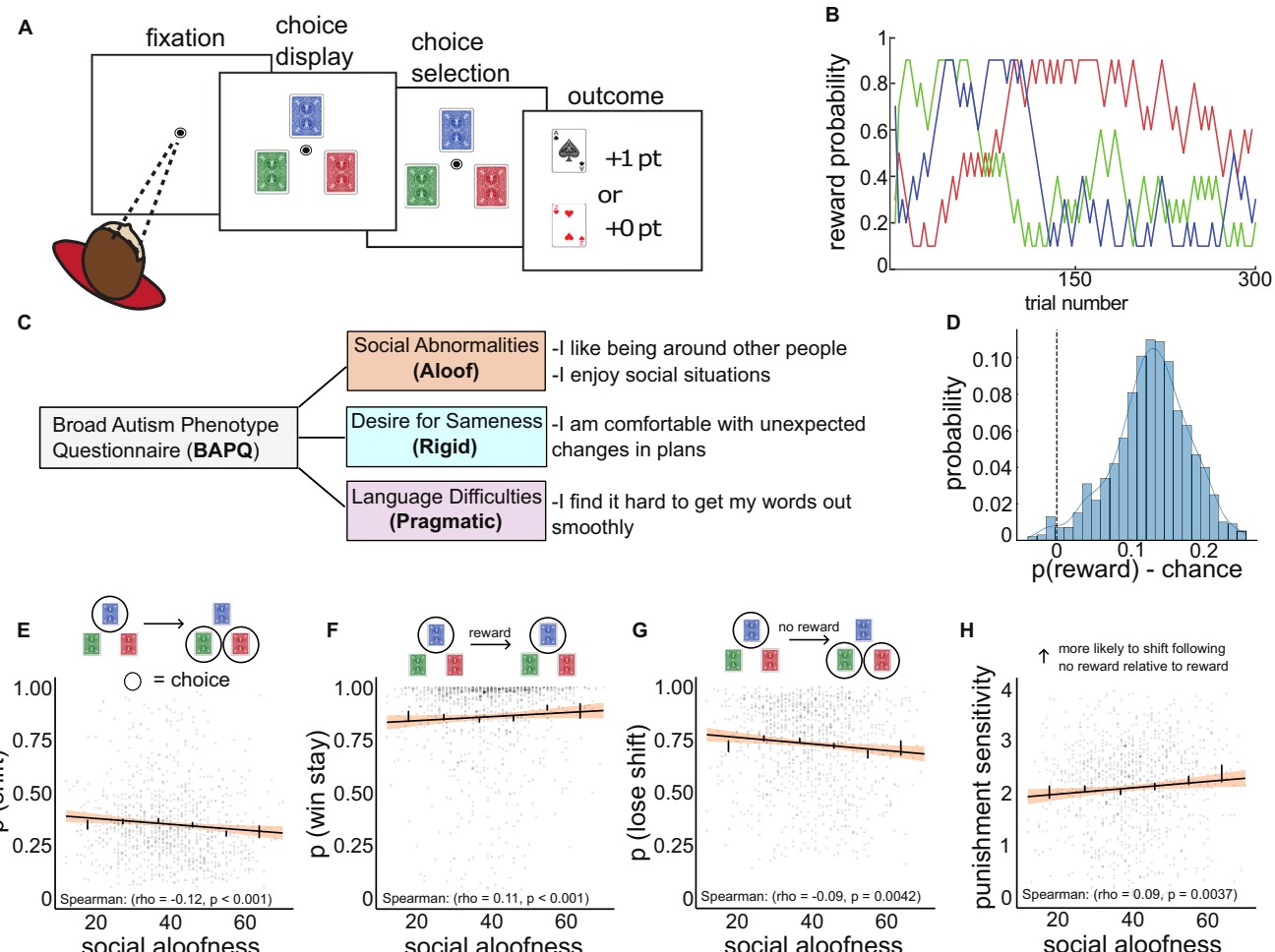

**Fig. 1 | Aloof subscale reflects changes in shift behavior. A** Schematic of the three-armed bandit task. **B** Example of the changing reward contingencies across 300 trials. **C** The broad autism phenotype questionnaire (BAPQ) was used to assess ASD like behaviors. The BAPQ contains three subscales: aloof, rigid, and pragmatic language, with example questions shown. **D** Probability of obtaining a reward relative to chance across participants. The dotted line represents chance. **E** Probability of shifting on a given trial relative to aloof subscale. **F** Probability of repeating a choice following reward relative to aloof subscale. **G** Probability of shifting from the previous choice following an unrewarded choice relative to the aloof subscale. **H** Sensitivity to unrewarded trials relative to rewarded trials, normalized to overall shiftiness of the participant, relative to aloof subscale. Error bars represent SEM for each bin, shaded areas represent 95% confidence interval for the regression estimate, $n = 1001$ for all analyses.

etc. The results revealed that individuals with higher aloofness were less likely to continue exploring after exploring in a previous trial, indicating shorter bouts of exploration (Spearman correlation (df = 999): rho = −0.14, 95% CI = [−0.200, −0.077], $p < 0.001$) (Fig. 2C). However, they were not increasing the length of their exploitative bouts (Spearman correlation (df = 999): rho = 0.05, 95% CI = [−0.007, 0.115], $p = 0.091$) (Fig. 2D). This suggests that the decrease in exploratory behaviors in high aloofness individuals is a product of their committing to a specific option more rapidly but not sticking to that option for a longer period compared to less aloof individuals.

## High aloofness is associated with reduced learning rate, lower decision noise, and increased choice stickiness

Previous studies, including our own, have demonstrated that changes in exploratory strategies can arise from multiple cognitive processes that can be captured by reinforcement learning (RL) parameters[4,5,29,30]. Specifically, exploratory behavior can be influenced by decision noise, which governs the randomness of choices, as well as by learning rates, which determine how strongly past outcomes influence future decisions. To better characterize the mechanisms underlying explore–exploit behaviors in relation to social aloofness, we fit several RL models to the choice data. We compared a base

RL model to an RL model with a choice kernel term (RLCK) (Fig. 2F), which accounts for choice biases and perseveration effects. The RLCK model provided a substantially better fit ($\Delta$AIC = 14,818.6), suggesting that, beyond reward learning, choice repetition independent of outcome is an important feature of the behavioral patterns demonstrated in this study. This model is especially useful for capturing the exploitative behavior observed in our HMM analysis, which revealed shorter exploratory bouts and persistent exploitation in high aloofness individuals. These patterns suggest that some of the exploitative behavior may arise not purely from stable reward expectations, but from increased choice perseveration.

Several key findings emerged from our RLCK analysis. First, we observed a trend-level relationship between aloofness and lower learning rates (Spearman correlation (df = 999): rho = −0.05, 95% CI = [−0.115, 0.009], $p = 0.096$) (Fig. 2E), suggesting that individuals higher in aloofness may update their reward expectations more slowly. Slower learning could reinforce exploitative behavior by reducing responsiveness to changes in reward contingencies. Second, higher aloof scores were significantly correlated with lower decision noise (Spearman correlation (df = 999): rho = 0.07, 95% CI = [0.007, 0.137], $p = 0.024$) (Fig. 2F), indicating a reduced tendency for stochastic or exploratory choices. This finding aligns with our HMM results, which showed fewer and shorter exploratory bouts in high

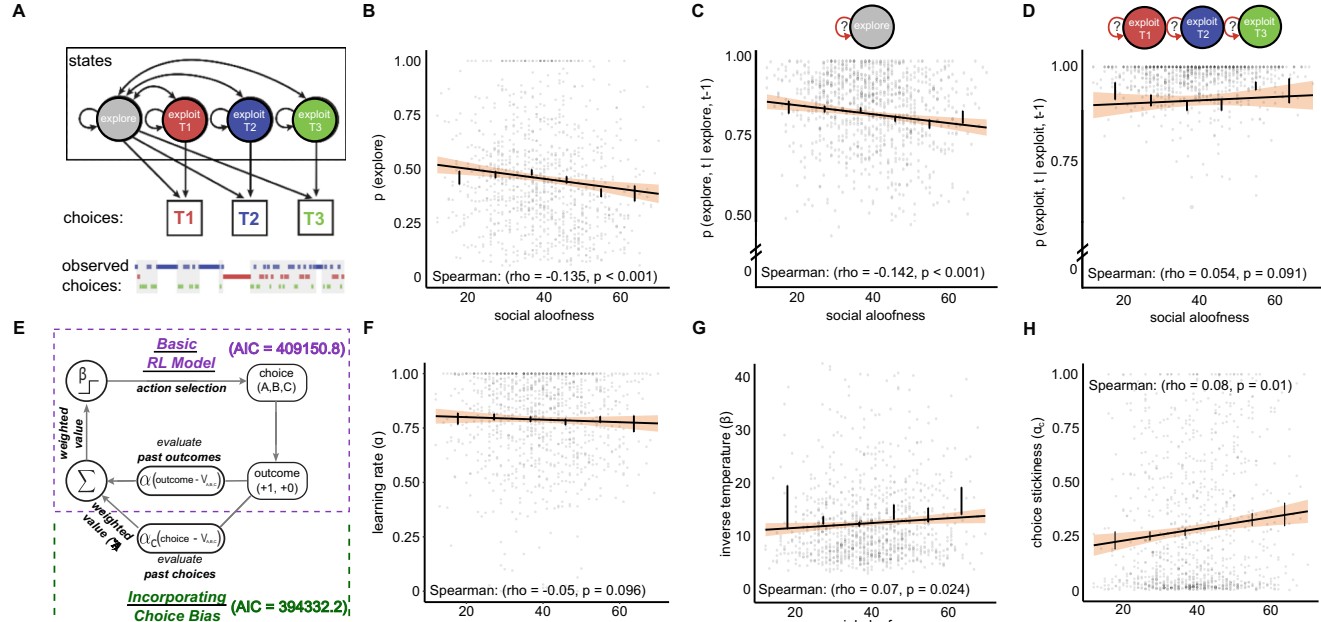

**Fig. 2 | High aloof individuals exhibit reduced exploration, increased choice repetition, and altered learning dynamics. A** Schematic of the hidden Markov model (HMM) used to label latent goal states underlying choice behaviors. The model incorporates three exploit labels for repetitions of each of the three choices, as well as an explore state in which choices were distributed across multiple options. Gray bars on observed choices represent choices labeled as exploratory. **B** Percentage of trials labeled as exploratory relative to aloof score. **C** Probability of exploring following an explore trial relative to aloof score. **D** Probability of exploiting following an exploit trial relative to aloof score. **E** Schematic of reinforcement learning model

(RL) and RL choice kernel (RLCK) models tested. AIC is included to show relative model performance when including the choice history kernel. **F** Learning rate as measured by the RLCK model relative to aloof score. **G** Inverse temperature as measured by the RLCK model relative to aloof score. Higher inverse temperature indicates lower decision noise. **H** Choice stickiness as measured by the RLCK model relative to aloof score. Higher values indicate a stronger tendency to repeat previous choices regardless of reward. Error bars represent SEM for each bin, shaded areas represent 95% confidence interval for the regression estimate, $n = 1001$ for all analyses.

aloof individuals, further supporting the conclusion that social aloofness is associated with reduced exploratory behavior.

A third result emerged when examining the choice kernel learning rate: higher aloofness was also significantly associated with increased choice stickiness (Spearman correlation (df = 999): rho = 0.08, 95% CI = [0.019, 0.141], $p = 0.010$) (Fig. 2H). This suggests that individuals with higher aloof scores are more likely to repeat previous choices regardless of reward history. Together, these results indicate that reduced exploration in socially aloof individuals arises from a combination of reduced decision noise and increased choice perseveration. Rather than simply being rigid or unresponsive to feedback, these individuals may be defaulting to habitual or exploitative strategies that favor repetition over adaptation. These findings highlight the utility of RL models incorporating choice history in revealing the cognitive mechanisms that underlie variation in both social and nonsocial flexibility.

### Social behavior-related questions show highest influence on exploratory strategy
Our correlation analyses revealed how differences in exploratory strategy are reflected by the subscales of BAPQ, specifically BAPQ aloof. Next we ask whether this relationship is driven by specific questions within the aloof subscale or social behavior related questions as a whole. Specifically, we adopted sparse canonical correlation analysis (sCCA), an unsupervised learning algorithm used to maximize the correlations between two datasets, to estimate the predictive power of each question.

The two input datasets consisted of 36 single-item scores from BAPQ (one for each question) and nine task parameters that best characterized cognitive strategy (Fig. 3A).

Chosen task parameters were (1) probability of exploration, (2) exploration potential, (3) punishment sensitivity, (4) probability of shifting away from the previous choice, (5) probability of repeating a rewarded

choice (win-stay), (6) probability of shifting away from a non-rewarded choice (lose-shift), (7) relative response time difference (*shift* vs *stay*), (8) averaged response time, and (9) probability of obtaining reward relative to chance.

We selected the first canonical variate based on the degree of covariance explained for further analysis. We performed a permutation to test for significance[35,36]. Null distribution was built by randomly re-assigning subjects' single-item scores to shuffle the original correlation between the two datasets. Results showed that the first canonical variate is significant (permutation, 5000 times, $p = 0.033$). To explain the psychiatric meaning of this component, we extracted canonical loadings for all single items within BAPQ (Fig. S1).

Interestingly, the component was dominated by symptoms related to social behaviors, with high-loading questions such as "I feel like I am really connecting with other people" (Fig. 3B). On the task side, this component was dominated by the probability of exploration. Further, we conducted a correlation analysis for p(explore) and the total score of these four symptoms, finding a strong correlation between the two (rho = −0.130, 95% CI = [−0.1905, −0.0687], $p < 0.001$) (Fig. 3D).

These results collectively indicate a connection between social flexibility, exploratory behaviors, and information-seeking strategies in decision-making tasks. Additionally, these findings showcase the exciting potential for computational models to provide insights into social behaviors, even in a fundamentally non-social task.

### Discussion
In this study, we examined how individual differences in self-reported social and non-social flexibility relate to decision-making strategies, specifically whether these traits manifest in measurable changes in explore–exploit behavior. Using computational modeling, we found that increased social aloofness was associated with decreased shift behavior, heightened

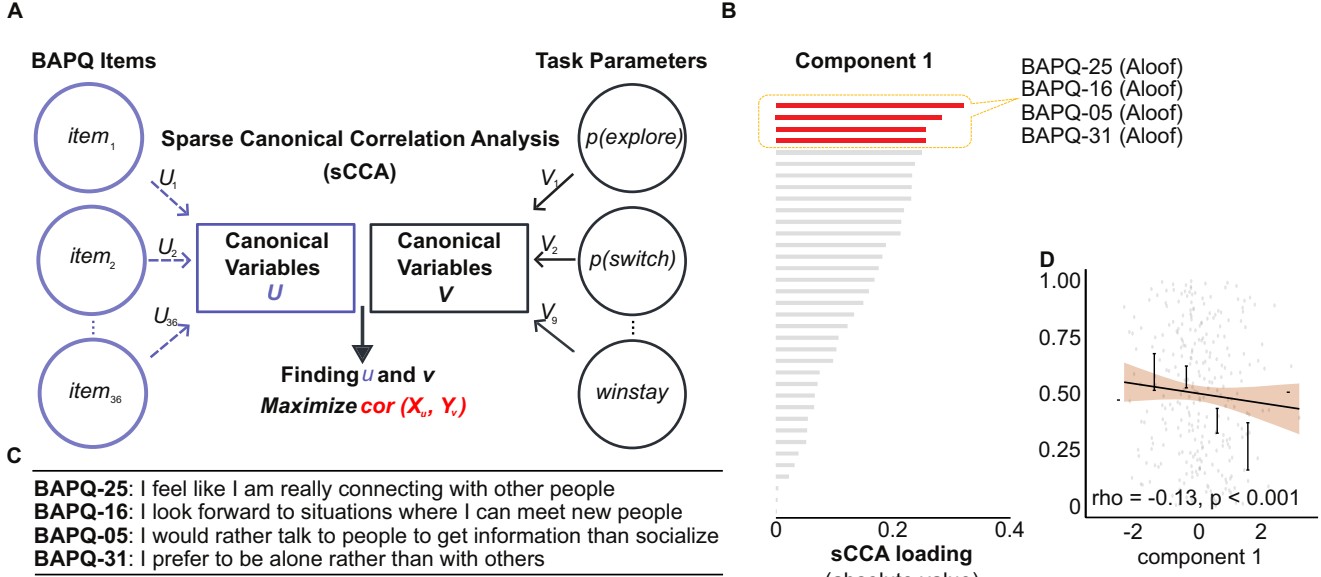

**Fig. 3 | Social specific questions show the highest influence on exploratory behaviors in sparse canonical correlation analysis. A** Schematic of sCCA used to examine the effects of specific questions on choice behavior and cognitive strategy. **B** Component one shows the highest loading for aloof questions, specifically. **C** Questions with the highest loading in component 1, listed in order of highest to lowest. **D** Overall probability of exploration relative to component 1. Error bars represent SEM for each bin, shaded areas represent 95% confidence interval for the regression estimate, n = 1001 for all analyses.

sensitivity to unrewarded trials, and reduced exploratory tendencies. Surprisingly, behavioral rigidity as measured by the BAPQ (a construct theoretically linked to cognitive inflexibility) did not significantly reflect changes in explore-exploit balance (Spearman correlation (df = 999): rho = 0.00, 95% CI = [−0.066, 0.059], p = 0.886), suggesting that these tasks may capture specific elements of behavioral and social flexibility that are not fully explained by cognitive rigidity.

Importantly, these effects were consistent across individual items within the aloof subscale, rather than isolated outlier responses, as revealed by our sparse canonical correlation analysis (sCCA). While social motivation and social skill are often interrelated, the aloof subscale of the BAPQ is generally understood to reflect an individual's interest in and preference for social engagement, rather than their competence in navigating social situations. This distinction is important for interpreting our findings. The association between aloofness and reduced exploration likely reflects motivational disengagement rather than impaired social cognition. However, reduced enjoyment of social interaction could still result from difficulty interpreting social cues, and future work incorporating behavioral or cognitive measures of social skill will be essential to further disentangle these components.

The observed aloofness-driven decrease in exploratory behavior could be influenced by both reward-related and reward-unrelated processes[40,41], and our findings provide evidence for both. Specifically, we found that social aloofness was associated with shorter bouts of exploration, but not longer bouts of exploitation, regardless of reward history. One possible explanation is that individuals with higher aloofness may find exploration less tolerable, potentially due to discomfort with uncertainty or novelty. Alternatively, they may find exploitation, or the reliance on known outcomes, more inherently satisfying or rewarding.

To further unpack these patterns and identify what might be driving the observed shift toward exploitative behavior, we examined trial-by-trial decision dynamics using reinforcement learning (RL) models. Our RL results revealed that reduced exploration in individuals high in aloofness is not due to generalized cognitive rigidity, but rather a combination of distinct decision-making tendencies. Specifically, aloofness was associated with lower decision noise, indicating a decreased tendency for stochastic or random exploration. We also observed a trend-level relationship between aloofness and lower learning rates, suggesting that high aloof individuals

may update their reward expectations more slowly, reinforcing a preference for exploitation over exploration. Importantly, aloofness was also positively correlated with the choice kernel learning rate ($\alpha_{CK}$), reflecting increased choice stickiness, or a stronger tendency to repeat prior choices regardless of reward. This finding indicates that more aloof individuals are not only less likely to try new options but are also more likely to default to recent action history as a guiding principle in decision-making. Together, these results align with our HMM findings and suggest that aloofness-linked reductions in exploration are driven by a convergence of lower decision noise, slower learning, and greater habitual repetition, rather than inflexible value updating alone.

Our finding that the aloof subscale was the most indicative of behavioral differences prompts the question: Is this due to actual variations in social flexibility, or could it be attributed to differences in self-reporting accuracy? An advantage of pairing a self-report questionnaire, such as the BAPQ, with a decision-making task, such as the restless bandit, is the ability to examine if what people *report* as their behavior matches what is *revealed* in a task assessing this behavior. To this point, it's interesting that self-reported levels of behavioral rigidity were not reflected in the behavioral strategy in the bandit task, with the rigid subscale showing no statistically significant correlation with common measures of behavioral flexibility such as shiftiness (Spearman correlation (df = 999): rho = 0.01, 95% CI = [−0.052, 0.071], p = 0.789). Self-reports are not necessarily accurate with regard to cognitive phenomena, and similar labels on self-report items and task measures do not mean that they access the same construct[42]. It is possible that participants are better able to accurately self-reflect on aloof subscale prompts, because they access the motivation to perform social behaviors, rather than mere competence[42,43]. Responding to "I enjoy chatting with people" (from the aloof subscale) better reflects motivation and preference than responding to questions such as "I can tell when it is time to change topics in conversation" (from the pragmatic subscale), reflecting competence. Because autistic traits have repeatedly been linked with changes in motivation[22,44,45], this may be an important contributor to why the aloof scale in a general population best assesses nonsocial motivation.

It is also possible that these findings are a result of the shared striatal circuitry of social information and reward processing[10,11,46,47]. Deficits in social behaviors could be a result of dysfunction in the same circuitry responsible for general reward and goal-directed behaviors and would

explain our observed correlations between aloofness, or lack of social motivation, and decision strategy, reflecting general motivation in this task. This interpretation would be consistent with a generalization of the social motivation hypothesis that has been proposed to explain the co-occurrence of changes in social and non-social reward sensitivity[48].

Prior research suggests that exploratory behavior in the restless bandit task may reflect a different domain of flexibility than what is typically assessed by the broad autism phenotype. Flexibility occurs on different timescales, from trial-to-trial adjustments to broader trait-level cognitive patterns across development[49]. In our study, exploratory behaviors correlated with social aloofness but were not significantly correlated with behavioral rigidity, suggesting that moment-to-moment decision-making flexibility may be more closely linked to motivation, specifically, the motivation to seek information, rather than a general resistance to change. This aligns with prior work suggesting that social engagement modulates exploration and learning[50,51] and that motivational factors influence uncertainty-driven decision-making[52,53]. The relationship between aloofness and reduced exploration may reflect a broader disengagement from novelty, independent of whether it is social or environmental in nature.

This raises the question of whether these results would persist across different task designs. The explore-exploit tradeoff has been extensively studied across various decision-making paradigms, each providing unique insights into cognitive flexibility. The horizon task[54] distinguishes between directed and random exploration by manipulating available information, while the novelty bandit task[55,56] assesses responses to entirely new options, capturing novelty-seeking tendencies. The restless bandit task used in our study requires continuous adaptation to dynamic reward contingencies, emphasizing real-time flexibility rather than broader trait-level patterns. This feature makes it particularly useful for studying how individuals balance reward exploitation with the need for ongoing information-seeking[4,5,7,40,41]. Given that our findings suggest a link between social aloofness and reduced exploration, it is possible that similar effects could emerge in other uncertainty-driven tasks, particularly those that require flexible updating of reward expectations. However, tasks that emphasize novelty-seeking, such as the novelty bandit task, may reveal additional nuances regarding whether social aloofness specifically dampens responses to new opportunities or whether it primarily affects how individuals adapt to shifting reward contingencies.

However, because different explore–exploit paradigms emphasize distinct cognitive and motivational processes, testing these effects across multiple tasks is critical for determining whether they generalize beyond the specific demands of the restless bandit task. An additional consideration is that while these types of tasks capture adaptive learning, they do not directly engage social cognition. Explicitly social decision-making paradigms, such as trust games[57] or multi-agent explore–exploit tasks[58,59], could test whether reduced exploration in highly aloof individuals is, in fact, reflected in social contexts. If similar patterns emerge in socially interactive tasks, this would suggest that the observed relationships reflect a domain-general reduction in information-seeking, with aloof individuals avoiding both social and non-social uncertainty. Conversely, if aloofness is only reflecting exploration deficits in non-social tasks, this would indicate a more domain-specific reduction in learning flexibility that does not translate to explicit social decision-making. These possibilities highlight the importance of cross-task comparisons in differentiating motivational versus cognitive contributions to exploration strategies.

## Limitations

An important next step is to determine whether the observed relationship between social aloofness and exploration is causal. Specifically, whether changes in social motivation can drive shifts in explore–exploit behavior. Longitudinal studies or experimental manipulations of social preference could help clarify this question. Additionally, it is important to consider the broader context in which these data were collected, as the COVID-19 pandemic significantly altered opportunities for social interaction. Future research should examine whether these associations persist as social

conditions stabilize or whether they were influenced by temporary changes in social engagement during that period. Despite these limitations, our findings demonstrate that ostensibly non-social decision-making tasks can reveal meaningful individual differences in social processing tendencies. This highlights their potential utility not only in studying social cognition in humans but also in translational animal models, providing a framework for investigating the neurocognitive mechanisms underlying social behavior.

## Data availability

Data sufficient to replicate the results is available at, in accordance with the approved IRB protocol: https://github.com/evanknep/BAPQ_Explore_Exploit.

## Code availability

The R code used to generate figures and statistics reported in this study is publicly available at: https://github.com/evanknep/BAPQ_Explore_Exploit. This repository contains a script containing figure creation and statistical analysis functions used to create figures presented in the manuscript.

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

## Acknowledgements

This work was supported by NIMH R01 MH123661, NIMH P50 MH119569, NIMH R01 MH127607, Discovery Grant RGPIN-2020-05577, Research-Scholar Award Fonds de Recherche du Québec—Santé (295755), and a Jacobs Foundation Research Fellowship. The funders had no role in study design, data collection and analysis, decision to publish or preparation of the manuscript.

## Author contributions
Evan Knep contributed to conceptualization, methodology, formal analysis, writing of the original draft, and review and editing of the manuscript. Xinyuan Yan contributed to conceptualization, methodology, formal analysis, writing of the original draft, and review and editing of the manuscript. Cathy S. Chen contributed to conceptualization, methodology, and review and editing of the manuscript. Suma Jacob contributed to conceptualization, methodology, supervision, funding acquisition, and review and editing of the manuscript. David P. Darrow contributed to conceptualization, methodology, supervision, funding acquisition, and review and editing of the manuscript. R. Becket Ebitz contributed to conceptualization, methodology, supervision, funding acquisition, and review and editing of the manuscript. Nicola Grissom contributed to conceptualization, methodology, supervision, funding acquisition, and review and editing of the manuscript. Alexander B. Herman contributed to conceptualization, methodology, supervision, funding acquisition, and review and editing of the manuscript.

## Competing interests
The authors declare no competing interests.
