## [Transparent Peer Review file · Communications Psychology]

Social aloofness is associated with non-social explore-exploit decisions

Corresponding Author: Mr Evan Knep

Version 0:

Decision Letter:

Dear Mr Knep,

Thank you for your patience during the peer-review process. Your manuscript titled "Explore-exploit behaviors predict social phenotypes" has now been seen by 3 reviewers, and I include their comments at the end of this message. They find your work of interest but raised some important points. We are interested in the possibility of publishing your study in *Communications Psychology*, but would like to consider your responses to these concerns and assess a revised manuscript before we make a final decision on publication.

We therefore invite you to revise and resubmit your manuscript, along with a point-by-point response to the reviewers. Please highlight all changes in the manuscript text file.

Editorially, we consider each of the points raised by the reviewers. But in particular:

- The need to provide a framework that connects: (1) the task (2) the processes/mechanisms hypothesised to be involved in the task and (3) the relevance of these mechanisms to the study of social phenotypes. R3 asks what rationale there is for focusing on the subscale of the BAPQ measures social difficulties related to autistic traits? Connecting an underlying mechanism to this component in particular seems prudent. Please also consider how to robustly test such a framework. R2 recommends fitting the choice data using reinforcement learning computational models noting that win-stay-lose-shift as a measure is unlikely to be able to adequately capture the strategies and choice profiles participants deploy that are needed to connect questionnaire responses to an underlying mechanism. This approach would be a way to unpack the complexity of the explore-exploit tradeoff, test different models (which pit different potential mechanisms against one another) against each other and explore relationships between parameters that capture specific aspects of explore-exploit processes and specific components of the questionnaire responses.
- R1 and R3 each ask how generalisable the results are. R3 asks whether you might expect the same results if social cognition had been directly assessed in an experiment (vs questionnaires). R1 asks if the results would generalise using other explore-exploit paradigms (e.g., the horizon task, novelty bandit task, etc.). Relatedly, R2 asks for a more detailed review of the explore-exploit literature along with definitions of "explore" and "exploit". Please consider the differences in how these concepts are assayed by different tasks and what exactly the research question here is in relation to the existing literature.
- In addition, R3 asks for more details on the experimental paradigm (e.g., task conditions), task behavior (e.g., distributions of participants' choices etc.), model parameters (e.g., distributions), model metrics (showing how well the HMMs described participants behavior) and the sCCA.
- Please also make explicit the exploratory nature of this work (as per R3's request).

I am attaching an Editorial Requests Table that details critical reporting requirements for the revised manuscript. Please

attend to each item and ensure your manuscript is fully compliant. If your revised manuscript is not aligned with these requests on major issues, such as those concerning statistics, it may be returned to you for further revisions without re-review.

Please submit the following items:

- Revised manuscript
- Point-by-point response to the referees' comments
- Cover letter (as a separate document)
- [Nature Research Reporting Summary](https://www.nature.com/documents/nr-reporting-summary.zip)
- [Editorial Policy Checklist](https://www.nature.com/documents/nr-editorial-policy-checklist.pdf)
- Completed Editorial Request Table (attached).

via this link: Link Redacted .

Additional guidance is available in our style and formatting guide [Communications Psychology formatting guide](https://www.nature.com/documents/commpsychol-style-formatting-guide-accept.pdf).

Best regards,

Neil Garrett

Neil Garrett, PhD
Editorial Board Member
Communications Psychology
orcid.org/0000-0003-1440-472X

REVIEWER EXPERTISE:

Reviewer #1: Modelling explore/exploit decision making
Reviewer #2: Modelling explore/exploit decision making, Hidden Markov Models
Reviewer #3: Social dysfunction / Autism, CCA

REVIEWER REPORTS:

Reviewer #1 (Remarks to the Author):

This study examines the correlation between explore-exploit behavior in a restless three-armed bandit and ASD-like behaviors assessed using the BAPQ. The team includes leaders in the field of computational modeling of explore-exploit tradeoffs in a restless bandit context, and the implications of an observed association between computational modeling of explore-exploit in the lab and the broader ASD phenotype based on self-report assays represent a compelling and exciting contribution. Additionally, the sCCA approach itself appears to be a robust and relatively novel method for integrating laboratory and real-world measurements, and is a real strength of the paper.

While the authors thoughtfully consider the potential limitations of self-report and how this may have led to null effects related to self-reported rigidity, I wonder if the unique processes tapped by the restless bandit itself might also play a role. It seems equally likely that other explore-exploit paradigms (e.g., the horizon task, novelty bandit task, etc.) may be more (or less) well-associated with self-report measures like the BAPQ items analyzed here. A discussion of how various computational models and task-based assays of explore-exploit behavior could be compared and contrasted in future studies aiming to predict 'real-world' outcomes like those measured by the BAPQ would be a welcome addition.

Otherwise, I commend the authors on a well-conducted and clearly communicated study that I am excited to see added to the literature.

Reviewer #2 (Remarks to the Author):

In this paper, the authors investigate whether behaviour in a restless bandit task can be linked to social and non-social components of flexibility. Participants performed a three-armed bandit task and completed the Broad Autism Phenotype Questionnaire to assess flexibility-related measures.

The title and main research idea of the paper are both intriguing and promising. To the best of my knowledge, research using Reinforcement Learning tasks has seldom explored their connections to behaviours associated with the autism spectrum. The exploration-exploitation trade-off provides a meaningful and robust framework to examine decision-making, as it allows participants to dynamically adapt to changes in their environment and employ diverse strategies. However, I find that the paper ultimately falls short of delivering on its initial promises.

First, the paper lacks a clear motivation for using an armed bandit task to address the research question. What is the connection between the processes involved (or assumed) in such a decision-making task and their relevance to the study of social phenotypes? Furthermore, the research question is not clearly articulated.

The structure of the paper is problematic as well. Readers unfamiliar with the explore-exploit dilemma or similar learning tasks may struggle to follow the rest of the content. The authors fail to explain what exploratory and exploitative behaviour entail or how decades of research in this area have advanced our understanding of such tasks. Despite this, the term "explore-exploit behaviours" — which even experienced readers may find unclear without proper context—are mentioned frequently throughout the text.

Also, there is no attempt to explain the potential mechanism that might be responsible for the observed results or to connect the current results to findings in the explore-exploit literature.

The rather superficial coverage of the topic is evident in the results section, where choice shifts or alternations between consecutive trials are used, which usually present a simplistic and often uninformative aspect of exploratory behaviour. Decades of research in this area have demonstrated the richness of participants' strategies and choice profiles in such tasks which may not be captured by choice shifts. Computational cognitive models have identified various strategies used in these tasks, such as the distinction between directed versus random exploration, the influence of perseveration (or inertia), and the updating or learning of reward values, among others.

In sum, I am not sure whether the current paper presents a coherent story to convincingly answer the research question - the authors would need to structure the paper in such a way that provides a narrative for the empirical study, identifies and tests the potential mechanisms, and goes deeper into the exploration-exploitation trade-off using relevant reinforcement learning computational models. In fact, relating the results from the questionnaire to estimated models parameters from relevant models might give us a much more refined and interesting picture of the research question.

Reviewer #3 (Remarks to the Author):

The paper "Explore exploit behaviors predict social phenotypes" reports on an interesting study on individual differences in explore-exploit decisions based on a self-report measure of autistic traits. I do appreciate that the paper topic, linking decision making to social cognition, is timely, the paper is clearly written, and the large sample of participants. However, important details and rationale are missing from the paper in its current form. It also seems that there is no preregistration of any portion of the study so authors should mention the exploratory nature of the study and tone down the claims.

There is a very strong claim about the ties of explore exploit decision making and social cognition but only one subscale of the BAPQ was used by authors to establish social functioning and this primarily measures social difficulties related to autistic traits. Yet no rationale is provided for why exactly this particular measure was used and autistic traits in the sample and their relevance for the task are barely mentioned. A stronger conclusion could be drawn from directly assessing social cognition in an experiment.

The experimental paradigm also needs to be better described. Authors mainly point to previous publications but don't really define the task conditions. In the same vein, I would want to see descriptive analyses of task behavior (distributions of participants' choices etc.), distributions of model parameters, as well as model metrics of how well the HMMs described participants behavior.

The sCCA is performed on item level of the BAPQ and I am not sure how diagnostic individual items are provided that they are non-independent from each other and therefore part of subscales. I would want to see a thorough item analysis and also splitting their data up into random parts of exploration and replication samples to establish correlations.

Communications Psychology is committed to improving transparency in authorship. As part of our efforts in this direction, we are now requesting that all authors identified as 'corresponding author' create and link their Open Researcher and Contributor Identifier (ORCID) with their account on the Manuscript Tracking System prior to acceptance. ORCID helps the scientific community achieve unambiguous attribution of all scholarly contributions. You can create and link your ORCID from the home page of the Manuscript Tracking System by clicking on 'Modify my Springer Nature account' and following the instructions in the link below. Please also inform all co-authors that they can add their ORCID to their accounts and that they must do so prior to acceptance.

Version 1:

Decision Letter:

Dear Mr Knep,

Your manuscript titled "Explore-exploit behaviors predict social phenotypes" has now been seen by our reviewers, whose comments appear below. In light of their advice I am delighted to say that we are happy, in principle, to publish a suitably revised version in Communications Psychology.

We therefore invite you to revise your paper one last time to address the remaining concerns of our reviewers and a list of editorial requests. At the same time we ask that you edit your manuscript to comply with our format requirements and to maximise the accessibility and therefore the impact of your work.

EDITORIAL REQUESTS:

SUBMISSION INFORMATION:

OPEN ACCESS:

At acceptance, you will be provided with instructions for completing the open access licence agreement on behalf of all authors. This grants us the necessary permissions to publish your paper. Additionally, you will be asked to declare that all

required third party permissions have been obtained, and to provide billing information in order to pay the article-processing charge (APC).

* **DATA AVAILABILITY:**

All Communications Psychology manuscripts must include a section titled "Data Availability" at the end of the Methods section. More information on this policy, is available in the Editorial Requests Table and at <http://www.nature.com/authors/policies/data/data-availability-statements-data-citations.pdf>

Link Redacted

Best regards,

Jennifer Bellingtier

Jennifer Bellingtier, PhD
Senior Editor
Communications Psychology

REVIEWERS' EXPERTISE:

Reviewer #1: Modelling explore/exploit decision making
Reviewer #2: Modelling explore/exploit decision making, Hidden Markov Models
Reviewer #3: Social dysfunction / Autism, CCA

REVIEWERS' COMMENTS:

Reviewer #1 (Remarks to the Author):

The authors have comprehensively responded to the comments raised by myself and the other reviewers.

Reviewer #2 (Remarks to the Author):

The authors have addressed all of my concerns in the revision. This is a significantly improved version of the paper.

Reviewer #3 (Remarks to the Author):

Overall the manuscript is much improved. I do think that it could have a stronger motivation for why the broad autism phenotype was chosen. Is it because autism traits are linked to differences in social skills and flexibility? In other words are authors after understanding sub-threshold autism in terms of explore-exploit differences or are they more interested in a scale that teases out social and non-social cognition? It is still not entirely clear. The authors mention model comparison in the paper but I would want to see it featured in the results (figure) and also interpreting those comparisons and finally I am not sure about the one reference to social motivation in the paper. Authors state "Because autistic traits have repeatedly been linked with changes in motivation (Chevallier et al., 2012; Mundy, 1995; Schultz, 2012), this may be an important contributor to why the aloof scale in a general population best assesses nonsocial motivation." I think they want to say social

motivation instead but I am having issues with a clear distinction between social motivation and social skills because I do think that they are strongly connected and hard to tease apart. A lack of enjoyment can come from feeling overwhelmed by social cues, misunderstood etc.

Summary of Revisions

We would like to thank the reviewers for their thoughtful and constructive feedback. In response, we have made substantial revisions to the manuscript to improve clarity, strengthen the theoretical framing, and expand the depth of analysis. Specifically, we:

- **Clarified the motivation** for using the restless bandit task and more explicitly articulated the research questions and relevance to social and non-social flexibility.
- **Expanded the introduction** to better define explore-exploit behavior and provide greater accessibility for readers less familiar with these paradigms.
- **Reinforced the exploratory nature of the study** throughout the manuscript and adjusted claims accordingly.
- **Incorporated reinforcement learning (RL) modeling** results to complement our Hidden Markov Model and sCCA findings, deepening mechanistic interpretation of explore-exploit behavior.
- **Provided full descriptions of the experimental task and analytic procedures** in the Introduction and Methods sections.
- **Added descriptive behavioral analyses and model parameter distributions** as supplemental figures to enhance transparency and interpretability.
- **Clarified our use of the BAPQ subscales**, including supplemental analyses showing why the aloof subscale was the primary focus, and addressed the use of item-level sCCA with justification and a full item analysis.

We believe these revisions have significantly strengthened the manuscript and hope the changes satisfactorily address the reviewers' comments. Item by item comments to each reviewer are below.

Reviewer #1 (Remarks to the Author)

This study examines the correlation between explore-exploit behavior in a restless three-armed bandit and ASD-like behaviors assessed using the BAPQ. The team includes leaders in the field of computational modeling of explore-exploit tradeoffs in a restless bandit context, and the implications of an observed association between computational modeling of explore-exploit in the lab and the broader ASD phenotype based on self-report assays represent a compelling and exciting contribution. Additionally, the sCCA approach itself appears to be a robust and relatively novel method for integrating laboratory and real-world measurements, and is a real strength of the paper.

While the authors thoughtfully consider the potential limitations of self-report and how this may have led to null effects related to self-reported rigidity, I wonder if the unique processes tapped by the restless bandit itself might also play a role. It seems equally likely that other explore-exploit paradigms (e.g., the horizon task, novelty bandit task, etc.) may be more (or less) well-associated with self-report measures like the BAPQ items analyzed here. A discussion of how

various computational models and task-based assays of explore-exploit behavior could be compared and contrasted in future studies aiming to predict 'real-world' outcomes like those measured by the BAPQ would be a welcome addition.

Otherwise, I commend the authors on a well-conducted and clearly communicated study that I am excited to see added to the literature.

We would like to thank Reviewer #1 for their thoughtful and constructive feedback. We appreciate their recognition of the strengths of our approach and the contribution of linking computational modeling of explore-exploit behavior to ASD-related traits. In response to their comments, we have expanded the discussion to address how the restless bandit may differ from other explore-exploit paradigms (e.g., horizon task, novelty bandit), and to acknowledge that the specificity of our task limits the scope of our conclusions. We now explicitly note in the text that this methodological choice opens important avenues for future research comparing different task structures to better understand what aspects of social behavior and social flexibility are being captured here.

We also incorporated the reviewer's suggestion regarding alternative modeling approaches alongside similar comments from another reviewer. Specifically, we included results from reinforcement learning (RL) modeling in the results and discussion sections to help further interpret individual differences in explore-exploit behavior. This addition strengthens the mechanistic component of the paper and allows for a richer interpretation of how specific computational parameters may relate to real-world traits.

Expanding Discussion On Restless vs. Alternative Explore-Exploit Paradigms

Adjusted Discussion Text:

“Prior research suggests that exploratory behavior in the restless bandit task may reflect a different domain of flexibility than what is typically assessed by the broad autism phenotype. Flexibility occurs on different timescales, from trial-to-trial adjustments to broader trait-level cognitive patterns across development (Hollenstein et al., 2013). In our study, exploratory behaviors correlated with social aloofness but not behavioral rigidity, suggesting that moment-to-moment decision-making flexibility may be more closely linked to motivation, specifically, the motivation to seek information, rather than a general resistance to change. This aligns with prior work suggesting that social engagement modulates exploration and learning (Behrens et al., 2008; Kappes et al., 2018) and that motivational factors influence uncertainty-driven decision-making (Jepma et al., 2020; Nasser et al., 2017). The relationship between aloofness and reduced exploration may reflect a broader disengagement from novelty, independent of whether it is social or environmental in nature.

This raises the question of whether these results would persist across different tasks designs. The explore-exploit tradeoff has been extensively studied across various decision-making paradigms, each providing unique insights into cognitive flexibility. The horizon task (Wilson et

al., 2014) distinguishes between directed and random exploration by manipulating available information, while the novelty bandit task (Costa et al., 2014; Djamshidian et al., 2011) assesses responses to entirely new options, capturing novelty-seeking tendencies. The restless bandit task used in our study requires continuous adaptation to dynamic reward contingencies, emphasizing real-time flexibility rather than broader trait-level patterns. This feature makes it particularly useful for studying how individuals balance reward exploitation with the need for ongoing information-seeking (C. Chen et al., 2023; C. S. Chen, Knep, et al., 2021; Ebitz et al., 2019; Kaske et al., 2022; Wilson et al., 2021). Given that our findings suggest a link between social aloofness and reduced exploration, it is possible that similar effects could emerge in other uncertainty-driven tasks, particularly those that require flexible updating of reward expectations. However, tasks that emphasize novelty-seeking, such as the novelty bandit task, may reveal additional nuances regarding whether social aloofness specifically dampens responses to new opportunities or whether it primarily affects how individuals adapt to shifting reward contingencies.

However, because different explore-exploit paradigms emphasize distinct cognitive and motivational processes, testing these effects across multiple tasks is critical for determining whether they generalize beyond the specific demands of the restless bandit task. An additional consideration is that while these types of tasks capture adaptive learning, they do not directly engage social cognition. Explicitly social decision-making paradigms, such as trust games (King-Casas et al., 2005) or multi-agent explore-exploit tasks (Leonardos et al., 2021; Sun et al., 2019), could test whether reduced exploration in highly aloof individuals is in fact reflected in social contexts. If similar patterns emerge in socially interactive tasks, this would suggest that the observed relationships reflect a domain-general reduction in information-seeking, with aloof individuals avoiding both social and non-social uncertainty. Conversely, if aloofness is only predictive of exploration deficits in non-social tasks, this would indicate a more domain-specific reduction in learning flexibility that does not translate to explicit social decision-making. These possibilities highlight the importance of cross-task comparisons in differentiating motivational versus cognitive contributions to exploration strategies.”

Expanding Results and Discussion to incorporate additional modeling

Results:

“High Aloofness is Associated with Reduced Decision Noise and Learning Rate

Previous studies, including our own, have demonstrated that changes in exploratory strategies can arise from multiple cognitive processes that can be captured by reinforcement learning (RL) parameters. Specifically, exploratory behavior can be influenced by decision noise, which governs the randomness of choices, as well as by learning rates, which determine how strongly past outcomes influence future decisions. To better characterize the mechanisms underlying explore-exploit behaviors in relation to social aloofness, we fit several RL models to the choice data. We compared a base RL model to an RL model with a choice kernel term (RLCK), which

accounts for choice biases and perseveration effects. The RLCK model provided a better fit, suggesting that these factors were important in explaining behavior. Consequently, all RL-based results presented use this model.

Two key findings emerged. First, higher aloof scores were correlated with lower decision noise, indicating that reduced exploration in high aloof individuals may stem from decreased random exploration rather than increased cognitive rigidity (Spearman correlation, $\rho = 0.07$, $p = 0.024$) (Figure 2E). This aligns with our HMM analysis, which showed fewer and shorter exploratory bouts in high aloof individuals. Second, we observed a trend-level relationship between aloofness and lower learning rates (Spearman correlation, $\rho = -0.05$, $p = 0.096$) (Figure 2F), suggesting that high aloof individuals may update their reward expectations more slowly. This could reinforce exploitative behavior by reducing responsiveness to changing reward contingencies. Together, these results indicate that aloofness-related reductions in exploration are driven by lower decision noise and potentially slower learning, rather than generalized cognitive inflexibility. These findings highlight the value of RL-based approaches in disentangling the cognitive mechanisms underlying social and non-social flexibility.”

Discussion:

“The observed aloofness-driven decrease in exploratory behavior could be influenced by both reward-related and reward-unrelated processes (Ebitz et al., 2019; Wilson et al., 2021), and our findings provide evidence for both. Specifically, we found that social aloofness was associated with shorter bouts of exploration, but not longer bouts of exploitation, regardless of reward history. One possible explanation is that individuals with higher aloofness may find exploration less tolerable, potentially due to discomfort with uncertainty or novelty. Alternatively, they may find exploitation, or the reliance on known outcomes, more inherently satisfying or rewarding.

Our reinforcement learning (RL) results further support this interpretation. We found that aloofness was associated with lower decision noise, suggesting that reduced exploratory behavior is not due to increased cognitive rigidity but rather a decreased tendency for stochastic or random exploration. Additionally, we observed a trend-level relationship between aloofness and lower learning rates, indicating that more aloof individuals may update their reward expectations more slowly, reinforcing a preference for exploitation. This is consistent with our behavioral finding that aloofness was linked to less decision-switching after a loss, suggesting that high aloof individuals persist in a given behavior despite receiving negative feedback that would typically prompt others to switch. These findings suggest potentially distinct mechanisms through which social and non-social cognitive flexibility manifest, with aloofness-linked reductions in exploration being driven by both reduced random decision noise and a more stable, less reactive learning process. Each of these processes may be underpinned by different neural mechanisms (Daw et al., 2006), highlighting the importance of computational approaches in disentangling the cognitive foundations of explore-exploit behavior.”

Reviewer #2 (Remarks to the Author):

In this paper, the authors investigate whether behaviour in a restless bandit task can be linked to social and non-social components of flexibility. Participants performed a three-armed bandit task and completed the Broad Autism Phenotype Questionnaire to assess flexibility-related measures.

The title and main research idea of the paper are both intriguing and promising. To the best of my knowledge, research using Reinforcement Learning tasks has seldom explored their connections to behaviours associated with the autism spectrum. The exploration-exploitation trade-off provides a meaningful and robust framework to examine decision-making, as it allows participants to dynamically adapt to changes in their environment and employ diverse strategies. However, I find that the paper ultimately falls short of delivering on its initial promises.

First, the paper lacks a clear motivation for using an armed bandit task to address the research question. What is the connection between the processes involved (or assumed) in such a decision-making task and their relevance to the study of social phenotypes? Furthermore, the research question is not clearly articulated.

The structure of the paper is problematic as well. Readers unfamiliar with the explore-exploit dilemma or similar learning tasks may struggle to follow the rest of the content. The authors fail to explain what exploratory and exploitative behaviour entail or how decades of research in this area have advanced our understanding of such tasks. Despite this, the term “explore-exploit behaviours” — which even experienced readers may find unclear without proper context—are mentioned frequently throughout the text.

Also, there is no attempt to explain the potential mechanism that might be responsible for the observed results or to connect the current results to findings in the explore-exploit literature.

The rather superficial coverage of the topic is evident in the results section, where choice shifts or alternations between consecutive trials are used, which usually present a simplistic and often uninformative aspect of exploratory behaviour. Decades of research in this area have demonstrated the richness of participants' strategies and choice profiles in such tasks which may not be captured by choice shifts. Computational cognitive models have identified various strategies used in these tasks, such as the distinction between directed versus random exploration, the influence of perseveration (or inertia), and the updating or learning of reward values, among others.

In sum, I am not sure whether the current paper presents a coherent story to convincingly answer the research question - the authors would need to structure the paper in such a way that provides a narrative for the empirical study, identifies and tests the potential mechanisms, and goes deeper into the exploration-exploitation trade-off using relevant reinforcement learning computational models. In fact, relating the results from the questionnaire to estimated models

parameters from relevant models might give us a much more refined and interesting picture of the research question.

We would like to thank Reviewer #2 for their time and feedback. Their comments helped us identify areas where additional clarity and context were needed to ensure that the story comes together in a coherent and convincing manner, and highlighted flaws in our structure that we had failed to see in the earlier version of the manuscript. In response, we clarified the motivation for using the restless bandit task in the context of examining cognitive flexibility across social and non-social domains, and more explicitly articulated the central research questions and motivations for the work. In doing so, we also expanded the background section to provide more comprehensive information on the explore-exploit framework and task design, making the manuscript more accessible and interpretable for a broader audience. Additionally, we clarified the exploratory nature of the study to ensure that interpretations are appropriately framed and claims are measured.

To increase the depth of behavioral interpretation and strengthen the mechanistic connection to the behavioral data, we incorporated reinforcement learning modeling results alongside the existing Hidden Markov Model and sparse canonical correlation analysis. Key RL parameters were graphed against BAPQ Aloof scores and added to figure 2. Additionally, distributions of RL parameters for all participants were added as a supplemental figure. We believe these additions significantly enhance the paper and directly address the reviewer's important suggestions.

More thorough introduction of explore-exploit dilemma:

“The explore-exploit tradeoff is a fundamental component of adaptive behavior, determining whether an agent seeks novel information (exploration) or capitalizes on prior rewards (exploitation). This balance is critical for decision-making across diverse contexts, shaping how individuals flexibly navigate uncertainty (Blanchard & Cañamero, 2006; Xu et al., 2021). Explore-exploit behaviors are predominantly studied in non-social environments, where agents must balance exploration and exploitation in resource foraging, economic decisions, and learning strategies. However, these behaviors also play a crucial role in social interactions. In social contexts, individuals must decide whether to engage with new partners or rely on existing relationships, influencing trust formation, cooperation, and social network dynamics (Do et al., 2024). Expanding social connections may provide access to novel resources, whereas maintaining established relationships ensures stability.”

Clarify motivation for using restless bandit/Clearly state research question:

“Decision-making tasks, especially bandit tasks, have proven effective at measuring explore-exploit balance in laboratory and ecological settings (C. Chen et al., 2023; C. S. Chen, Knep, et al., 2021; Dong et al., 2016; Kaske et al., 2022; Laureiro-Martínez & Brusoni, 2018). Bandit tasks require participants to repeatedly choose between options with uncertain and often

changing reward probabilities, forcing them to balance the short-term benefits of exploiting known rewards with the potential long-term gains of exploring less familiar options. By tracking how individuals adapt their choices in response to changing reward contingencies, these tasks offer a computationally precise method for assessing decision-making flexibility. However, it remains unclear what aspects of real-world functioning are accessed by laboratory explore-exploit measures such as these. As these tasks typically involve sequences of repetitive choices in a single-agent setting, they are often argued to primarily reflect cognitive flexibility or rigidity, potentially failing to capture the social flexibility component of explore-exploit behavior.

Yet these tasks may go beyond measuring surface-level cognitive traits, with evidence suggesting that they often tap into deeper neurocomputational mechanisms that underlie more complex behaviors (Hogeveen et al., 2022). For example, it has been long understood that the same underlying cognitive mechanisms that facilitate flexible decision-making in non-social contexts also play a crucial role in social interactions (Báez-Mendoza & Schultz, 2013; Bhanji & Delgado, 2014). If we can leverage explore-exploit paradigms to access social flexibility, we can unlock new insights into the foundational cognitive processes driving both social and non-social flexibility, offering a more comprehensive understanding of adaptive behavior in complex environments (Albein-Urios et al., 2018; Hollocks et al., 2023; Lei et al., 2022; Piche et al., 2018; Uddin, 2021; Wolff et al., 2023).”

Connect findings to mechanisms:

“Previous studies, including our own, have demonstrated that changes in exploratory strategies can arise from multiple cognitive processes that can be captured by reinforcement learning (RL) parameters. Specifically, exploratory behavior can be influenced by decision noise, which governs the randomness of choices, as well as by learning rates, which determine how strongly past outcomes influence future decisions. To better characterize the mechanisms underlying explore-exploit behaviors in relation to social aloofness, we fit several RL models to the choice data. We compared a base RL model to an RL model with a choice kernel term (RLCK), which accounts for choice biases and perseveration effects. The RLCK model provided a better fit, suggesting that these factors were important in explaining behavior. Consequently, all RL-based results presented use this model.

Two key findings emerged. First, higher aloof scores were correlated with lower decision noise, indicating that reduced exploration in high aloof individuals may stem from decreased random exploration rather than increased cognitive rigidity (Spearman correlation, $\rho = 0.07$, $p = 0.024$) (Figure 2E). This aligns with our HMM analysis, which showed fewer and shorter exploratory bouts in high aloof individuals. Second, we observed a trend-level relationship between aloofness and lower learning rates (Spearman correlation, $\rho = -0.05$, $p = 0.096$) (Figure 2F), suggesting that high aloof individuals may update their reward expectations more slowly. This could reinforce exploitative behavior by reducing responsiveness to changing reward contingencies. Together, these results indicate that aloofness-related reductions in exploration are driven by lower decision noise and potentially slower learning, rather than

generalized cognitive inflexibility. These findings highlight the value of RL-based approaches in disentangling the cognitive mechanisms underlying social and non-social flexibility.”

Deepen model interpretation:

“Importantly, these effects were consistently predicted by items within the aloof subscale, rather than isolated outlier responses, as revealed by our sparse canonical correlation analysis (sCCA). Furthermore, RL modeling indicated that reduced exploration in high aloof individuals was primarily driven by lower decision noise rather than increased perseveration or inflexibility. These findings suggest that perceived social flexibility may reflect broader cognitive tendencies that extend to non-social decision-making processes. By linking social disengagement to reduced exploration, our results highlight the potential of computational modeling in uncovering cognitive mechanisms underlying both social and non-social adaptability.

The observed aloofness-driven decrease in exploratory behavior could be influenced by both reward-related and reward-unrelated processes (Ebitz et al., 2019; Wilson et al., 2021), and our findings provide evidence for both. Specifically, we found that social aloofness was associated with shorter bouts of exploration, but not longer bouts of exploitation, regardless of reward history. One possible explanation is that individuals with higher aloofness may find exploration less tolerable, potentially due to discomfort with uncertainty or novelty. Alternatively, they may find exploitation, or the reliance on known outcomes, more inherently satisfying or rewarding.

Our reinforcement learning (RL) results further support this interpretation. We found that aloofness was associated with lower decision noise, suggesting that reduced exploratory behavior is not due to increased cognitive rigidity but rather a decreased tendency for stochastic or random exploration. Additionally, we observed a trend-level relationship between aloofness and lower learning rates, indicating that more aloof individuals may update their reward expectations more slowly, reinforcing a preference for exploitation. This is consistent with our behavioral finding that aloofness was linked to less decision-switching after a loss, suggesting that high aloof individuals persist in a given behavior despite receiving negative feedback that would typically prompt others to switch. These findings suggest potentially distinct mechanisms through which social and non-social cognitive flexibility manifest, with aloofness-linked reductions in exploration being driven by both reduced random decision noise and a more stable, less reactive learning process. Each of these processes may be underpinned by different neural mechanisms (Daw et al., 2006), highlighting the importance of computational approaches in disentangling the cognitive foundations of explore-exploit behavior.”

Reviewer #3 (Remarks to the Author):

The paper “Explore exploit behaviors predict social phenotypes” reports on an interesting study on individual differences in explore-exploit decisions based on a self-report measure of autistic traits. I do appreciate that the paper topic, linking decision making to social cognition, is timely, the paper is clearly written, and the large sample of participants. However, important details and rationale are missing from the paper in its current form. It also seems that there is no preregistration of any portion of the study so authors should mention the exploratory nature of the study and tone down the claims.

There is a very strong claim about the ties of explore exploit decision making and social cognition but only one subscale of the BAPQ was used by authors to establish social functioning and this primarily measures social difficulties related to autistic traits. Yet no rationale is provided for why exactly this particular measure was used and autistic traits in the sample and their relevance for the task are barely mentioned. A stronger conclusion could be drawn from directly assessing social cognition in an experiment.

The experimental paradigm also needs to be better described. Authors mainly point to previous publications but don't really define the task conditions. In the same vein, I would want to see descriptive analyses of task behavior (distributions of participants' choices etc.), distributions of model parameters, as well as model metrics of how well the HMMs described participants behavior.

The sCCA is performed on item level of the BAPQ and I am not sure how diagnostic individual items are provided that they are non-independent from each other and therefore part of subscales. I would want to see a thorough item analysis and also splitting their data up into random parts of exploration and replication samples to establish correlations.

We appreciate Reviewer #3's careful reading of the manuscript and their pointed suggestions, which helped us refine both the structure and clarity of our study. Their feedback was particularly valuable in identifying the need for a more clearly articulated rationale for the task design and a stronger introduction to the relevance of the BAPQ subscales. In response, we have expanded the introduction to better motivate the use of the restless bandit task for examining cognitive flexibility and clarified how each BAPQ subscale relates to our research aims. We now more explicitly explain the focus on the aloof subscale by noting that all three subscales were analyzed, but only aloofness showed consistent associations with behavior. To support this clarification, we have added supplemental figures displaying behavioral metrics plotted against the pragmatic and rigid subscales, and expanded the introduction to provide greater context for each subscale's theoretical grounding. Additionally, we have reinforced the exploratory nature of the study throughout the manuscript to ensure that interpretations and claims are appropriately measured.

We also addressed the need for greater clarity and depth in the description of the experimental paradigm and behavioral analyses. The task design is now more fully described in both the Introduction and Methods sections. Descriptive analyses of task behavior—including choice

distributions across all participants, as well as distributions of key behavioral variables and model parameters—have been added as a supplemental figure. In addition, we now include a detailed individual item analysis of all 36 BAPQ items used in the sCCA, presented in the supplement to enhance transparency and interpretability. This analysis allows readers to assess the contribution of each item to the multivariate solution and evaluate whether patterns are driven by specific items or clusters of related content. While we acknowledge that BAPQ items are not fully independent due to their organization into subscales, we used a sparse CCA approach to help mitigate issues of multicollinearity by selecting only the most relevant subset of features. By providing the full item-level results, we aim to support a more nuanced interpretation and allow for clearer mapping between behavioral patterns and specific facets of autistic traits.

Acknowledging Exploratory Nature

“Abstract: Given prior links between exploratory behavior and cognitive rigidity, we were surprised to find that differences in choice behavior and exploration were in fact most predictive of social phenotypes as captured by the BAPQ aloof subscale.

Introduction: By exploring these relationships, we aim to determine whether explore-exploit laboratory paradigms, such as the restless bandit task, capture both social and non-social flexibility. If both aloofness and rigidity predict a more exploitative strategy, it would suggest that these tasks reflect a broader profile of cognitive and social inflexibility, with shared mechanisms driving disengagement from novelty across domains. Alternatively, if these traits show distinct relationships with explore-exploit behavior, it may indicate that social and non-social flexibility are supported by separate processes, with exploration in decision-making reflecting different underlying motivations depending on the context. Understanding these connections will help clarify whether laboratory paradigms measuring explore-exploit tradeoffs provide meaningful insights into flexibility beyond non-social decision-making, offering a more comprehensive framework for studying adaptive behavior in dynamic environments.

Surprisingly, we found the only subscale predictive of explore-exploit behavior was social aloofness, while no significant relationship emerged with behavioral rigidity or pragmatic language difficulties. This challenges the assumption that explore-exploit behavior primarily reflects non-social cognitive flexibility, instead suggesting that the motivational and engagement-related aspects of decision-making may be more relevant. Individuals higher in aloofness, who tend to disengage from social interactions, also exhibited reduced exploratory behavior in the bandit task, potentially reflecting a broader pattern of reduced motivation to seek new information across both social and non-social domains. This finding highlights the potential for laboratory-based decision-making tasks to capture aspects of social flexibility, raising new questions about how social motivation shapes exploratory decision-making in uncertain environments. Further research is needed to disentangle whether aloofness-driven reductions in exploration stem from a general disengagement from novel opportunities or a more domain-specific social cognitive mechanism.

Discussion: In this study, we examined how individual differences in self-reported social and non-social flexibility relate to decision-making strategies, specifically whether these traits manifest in measurable changes in explore-exploit behavior. Using computational modeling, we found that increased social aloofness was associated with decreased shift behavior, heightened sensitivity to unrewarded trials, and reduced exploratory tendencies. Surprisingly, behavioral rigidity as measured by the BAPQ (a construct theoretically linked to cognitive inflexibility) did not predict explore-exploit behaviors, suggesting that these tasks may capture specific elements of behavioral and social flexibility that are not fully explained by cognitive rigidity.

Importantly, these effects were consistently predicted by items within the aloof subscale, rather than isolated outlier responses, as revealed by our sparse canonical correlation analysis (sCCA). Furthermore, RL modeling indicated that reduced exploration in high aloof individuals was primarily driven by lower decision noise rather than increased perseveration or inflexibility. These findings suggest that perceived social flexibility may reflect broader cognitive tendencies that extend to non-social decision-making processes. By linking social disengagement to reduced exploration, our results highlight the potential of computational modeling in uncovering cognitive mechanisms underlying both social and non-social adaptability.”

Expanding Introduction and Justification of Focus on Specific BAPQ Subscales

“In this study, we aimed to examine how explore-exploit balance in a restless bandit task relates to social and non-social flexibility in a large online sample (n=1001). To examine distinct components of flexibility, we used the Broad Autism Phenotype Questionnaire (BAPQ), a self-report survey designed to capture variation in social and non-social cognitive traits in a community sample (Keith et al., 2019; Maxwell et al., 2013; Sandercock et al., 2020; Seidman et al., 2012). The BAPQ includes three subscales: social aloofness, behavioral rigidity, and pragmatic language, each measuring different aspects of cognitive and social adaptability. Given our interest in how decision-making strategies reflect individual differences in flexibility, we focused on the aloof and rigid subscales, as both directly relate to patterns of engagement with dynamic environments. Importantly, these subscales are often correlated (Hurley et al., 2007), raising the question of whether individuals with a more exploitative decision-making style exhibit both high aloofness and rigidity, or if these traits predict distinct patterns of explore-exploit behavior.

The social aloofness subscale captures an individual’s tendency to withdraw or disengage from social interactions, making it highly relevant to social flexibility. Individuals high in aloofness often show reduced social motivation (Chevallier et al., 2012), which could influence exploration tendencies, particularly in uncertain or changing environments. If exploration in decision-making reflects an underlying drive for engagement with new opportunities, more aloof individuals may be less inclined to seek novel options, favoring a more exploitative approach that prioritizes known rewards. This aligns with findings that reduced social motivation is linked to behavioral patterns of social avoidance (Gable, 2006) and lower sensitivity to changing reward contingencies (Luckhardt et al., 2023). However, it remains unclear whether this tendency generalizes beyond social settings to influence exploration in non-social decision-making contexts.

The rigidity scale, on the other hand, measures inflexibility in thought and behavior, including resistance to change and a strong preference for routine. This subscale directly aligns with cognitive and behavioral adaptability in non-social contexts, making it particularly well-suited for testing relationships between decision-making flexibility and explore-exploit behaviors. A more rigid individual may struggle to adjust to changing reward contingencies, leading to a greater reliance on previously learned strategies rather than exploring new possibilities. However, given the established correlation between rigidity and aloofness (Flippin & Watson, 2018), it is possible that exploitative individuals exhibit both tendencies, with aloofness contributing to reduced engagement in novel experiences and rigidity reinforcing reliance on established routines.

By exploring these relationships, we aim to determine whether explore-exploit laboratory paradigms, such as the restless bandit task, capture both social and non-social flexibility. If both aloofness and rigidity predict a more exploitative strategy, it would suggest that these tasks reflect a broader profile of cognitive and social inflexibility, with shared mechanisms driving disengagement from novelty across domains. Alternatively, if these traits show distinct relationships with explore-exploit behavior, it may indicate that social and non-social flexibility are supported by separate processes, with exploration in decision-making reflecting different underlying motivations depending on the context. Understanding these connections will help clarify whether laboratory paradigms measuring explore-exploit tradeoffs provide meaningful insights into flexibility beyond non-social decision-making, offering a more comprehensive framework for studying adaptive behavior in dynamic environments.”

Figure S3. BAPQ Total, Shift, and Pragmatic Scores Fail to Predict Choice Behavior

A) [Column] Probability of shifting on a given trial relative to BAPQ total, rigid, and pragmatic scores. B) [Column] Probability of repeating a choice following reward relative to BAPQ total, rigid, and pragmatic scores. C) [Column] Probability of shifting from previous choice following an unrewarded choice relative to BAPQ total, rigid, and pragmatic scores. D) [Column] Sensitivity to unrewarded trials relative to rewarded trials, normalized to overall shiftiness of the participant, relative to BAPQ total, rigid, and pragmatic scores.

Expanded Description of Experimental Paradigm - Inclusion of Task Details

“Introduction: The explore-exploit tradeoff is a fundamental component of adaptive behavior, determining whether an agent seeks novel information (exploration) or capitalizes on prior rewards (exploitation). This balance is critical for decision-making across diverse contexts, shaping how individuals flexibly navigate uncertainty (Blanchard & Cañamero, 2006; Xu et al., 2021). Explore-exploit behaviors are predominantly studied in non-social environments, where agents must balance exploration and exploitation in resource foraging, economic decisions, and learning strategies. However, these behaviors also play a crucial role in social interactions. In social contexts, individuals must decide whether to engage with new partners or rely on existing relationships, influencing trust formation, cooperation, and social network dynamics (Do et al., 2024). Expanding social connections may provide access to novel resources, whereas maintaining established relationships ensures stability.

Decision-making tasks, especially bandit tasks, have proven effective at measuring explore-exploit balance in laboratory and ecological settings (C. Chen et al., 2023; C. S. Chen, Knep, et al., 2021; Dong et al., 2016; Kaske et al., 2022; Laureiro-Martínez & Brusoni, 2018). Bandit tasks require participants to repeatedly choose between options with uncertain and often changing reward probabilities, forcing them to balance the short-term benefits of exploiting known rewards with the potential long-term gains of exploring less familiar options. By tracking how individuals adapt their choices in response to changing reward contingencies, these tasks offer a computationally precise method for assessing decision-making flexibility. However, it remains unclear what aspects of real-world functioning are accessed by laboratory explore-exploit measures such as these. As these tasks typically involve sequences of repetitive choices in a single-agent setting, they are often argued to primarily reflect cognitive flexibility or rigidity, potentially failing to capture the social flexibility component of explore-exploit behavior.

Yet these tasks may go beyond measuring surface-level cognitive traits, with evidence suggesting that they often tap into deeper neurocomputational mechanisms that underlie more complex behaviors (Hogeveen et al., 2022). For example, it has been long understood that the same underlying cognitive mechanisms that facilitate flexible decision-making in non-social contexts also play a crucial role in social interactions (Báez-Mendoza & Schultz, 2013; Bhanji & Delgado, 2014). If we can leverage explore-exploit paradigms to access social flexibility, we can unlock new insights into the foundational cognitive processes driving both social and non-social flexibility, offering a more comprehensive understanding of adaptive behavior in complex environments (Albein-Urios et al., 2018; Hollocks et al., 2023; Lei et al., 2022; Piche et al., 2018; Uddin, 2021; Wolff et al., 2023).

Methods: Exploratory behaviors were measured using a three-arm restless bandit task (Liu et al., 2013; Speekenbrink & Konstantinidis, 2015). Each trial, participants were presented with three choices, each of which was associated with a reward probability that changed randomly and independently over time. Rewarded trials resulted in participants earning 1 point. The probabilistic nature of the task necessitates exploratory behaviors to monitor changes in reward rates as the task progresses, while encouraging exploitation of a target while the chance of reward is high. Participants were initially screened with a tutorial section, in which they needed to demonstrate basic task proficiency over 15 practice trials prior to beginning the final task. Each participant's restless "walk", or the volatility of the reward probability across 300 trials, was dictated by predetermined parameters for the likelihood of change in reward probability each trial (hazard rate) and the subsequent magnitude of change in reward probability (step size). In this experiment we used a hazard rate of 0.6667 and a step size of +/- 0.2, constrained within the range of [0.1-0.9]."

Descriptive Analyses of Task Behavior - Visualization of Distributions

Figure S2. Distributions of choice behavior and RL Parameters.

A) Density plot showing the distribution of choice proportions across the three options (left, center, right). **B)** Distribution of win-stay probability, or the likelihood of repeating a choice following a rewarded trial. **C)** Distribution of lose-shift probability, or the likelihood of switching choices following an unrewarded trial. **D)** Overall shift probability, representing the likelihood of changing choices on any given trial. **E)** Distribution of the learning rate (α), which determines how strongly past outcomes influence future choices. **F)** Distribution of the inverse temperature (β), which governs the degree of exploitation versus exploration in decision-making.

SCCA Item-By-Item Analysis

Figure S1: Sparse Canonical Correlation Analysis Reveals Behavioral Relevance of all 36 BAPQ Items

Left column contains the 36 questions within the BAPQ, with horizontal bars indicating the relative loading on a dataset consisting of nine behavioral indices: (1) $p(\text{explore})$, or percent of time spent in exploration, (2) exploration potential inferred from our hidden markov model, and the other six model-free indices included, (3) punishment sensitivity, (4) percentage of choices that were shifts, (5) $p(\text{win-stay})$, (6) $p(\text{lose-shift})$, (7) relative response time (shift vs stay), (8) averaged response time, (9) performance over chance level.

Response to Reviewer 3

We sincerely thank Reviewer 3 for their thoughtful comments. We appreciate the recognition of the manuscript's improvements from the previous revision, and are grateful for their suggestions. We think they resulted in clearer rationale for our measurement approach and theoretical interpretations.

Clarifying the Motivation for Using the Broad Autism Phenotype Questionnaire (BAPQ)

In response to the reviewer's questions about why we selected the BAPQ and whether our focus is on subthreshold autism or on differentiating social and non-social cognition, we have clarified this point in the introduction. Specifically, we now write:

“Importantly, although the BAPQ originates in autism research, our goal was not to study subclinical autism, but to leverage the BAPQ as a dimensional tool to parse social disengagement (aloofness) and cognitive inflexibility (rigidity) as potential drivers of explore-exploit behaviors in the broader population.”

Reinforcement Learning (RL) Model Interpretation and Visualization

In response to the request for a more complete presentation and interpretation of our modeling work, we made several important updates. We revised Figure 2 to include (1) a schematic of the base RL and RLCK models, (2) a comparison of model fit using Akaike Information Criterion (AIC), and (3) a new panel showing the association between the RLCK model's measure of choice stickiness and the BAPQ aloof subscale.

Along with the new schematic, we highlight an additional finding in figure 2:

“A third result emerged when examining the choice kernel learning rate: higher aloofness was also significantly associated with increased choice stickiness (Spearman correlation, $\rho = 0.08$, $p = 0.01$) (Figure 2H). This suggests that individuals with higher aloof scores are more likely to repeat previous choices regardless of reward history. Together, these results indicate that reduced exploration in socially aloof individuals arises from a combination of reduced decision noise and increased choice perseveration. Rather than simply being rigid or unresponsive to feedback, these individuals may be defaulting to habitual or exploitative strategies that favor repetition over adaptation. These findings highlight the utility of RL models incorporating choice

history in revealing the cognitive mechanisms that underlie variation in both social and non-social flexibility.”

We also expand the discussion to provide a more integrated interpretation of our HMM and RL results. We now include the following passage:

“To further unpack these patterns and identify what might be driving the observed shift toward exploitative behavior, we examined trial-by-trial decision dynamics using reinforcement learning (RL) models. Our RL results revealed that reduced exploration in individuals high in aloofness is not due to generalized cognitive rigidity, but rather a combination of distinct decision-making tendencies. Specifically, aloofness was associated with lower decision noise, indicating a decreased tendency for stochastic or random exploration. We also observed a trend-level relationship between aloofness and lower learning rates, suggesting that high aloof individuals may update their reward expectations more slowly, reinforcing a preference for exploitation over exploration. Importantly, aloofness was also positively correlated with the choice kernel learning rate (α_{CK}), reflecting increased choice stickiness, or a stronger tendency to repeat prior choices regardless of reward. This finding indicates that more aloof individuals are not only less likely to try new options, but are also more likely to default to recent action history as a guiding principle in decision-making. Together, these results align with our HMM findings and suggest that aloofness-linked reductions in exploration are driven by a convergence of lower decision noise, slower learning, and greater choice stickiness, rather than inflexible value updating alone.”

Clarifying The Distinction Between Social Motivation and Social Skills

We agree that it’s important to highlight the difference between social motivation and social skills. To clarify this point, we revised the discussion to be more explicit about how we are interpreting these results, while also acknowledging the nuanced nature of the experimental measure.

Importantly, these effects were consistently predicted by items within the aloof subscale, rather than isolated outlier responses, as revealed by our sparse canonical correlation analysis (sCCA). While social motivation and social skill are often interrelated, the aloof subscale of the BAPQ is generally understood to reflect an individual’s interest in and preference for social engagement, rather than their competence in navigating social situations. This distinction is important for interpreting our findings. The association between aloofness and reduced exploration likely reflects motivational disengagement rather than impaired social cognition. However, reduced enjoyment of social interaction

could still result from difficulty interpreting social cues, and future work incorporating behavioral or cognitive measures of social skill will be essential to further disentangle these components.